# A Novel Deep Reinforcement Learning (DRL) Algorithm to Apply Artificial Intelligence-Based Maintenance in Electrolysers

Abiodun Abiola *, Francisca Segura Manzano * and José Manuel Andújar

Reseaarch Centre on Technology, Energy and Sustainability (CITES), University of Huelva, Campus El Carmen, 21071 Huelva, Spain; andujar@diesia.uhu.es
* Correspondence: abiodunolatokunbo.abiola@alumni.urv.cat (A.A.); francisca.segura@diesia.uhu.es (F.S.M.)

**Abstract:** Hydrogen provides a clean source of energy that can be produced with the aid of electrolysers. For electrolysers to operate cost-effectively and safely, it is necessary to define an appropriate maintenance strategy. Predictive maintenance is one of such strategies but often relies on data from sensors which can also become faulty, resulting in false information. Consequently, maintenance will not be performed at the right time and failure will occur. To address this problem, the artificial intelligence concept is applied to make predictions on sensor readings based on data obtained from another instrument within the process. In this study, a novel algorithm is developed using Deep Reinforcement Learning (DRL) to select the best feature(s) among measured data of the electrolyser, which can best predict the target sensor data for predictive maintenance. The features are used as input into a type of deep neural network called long short-term memory (LSTM) to make predictions. The DLR developed has been compared with those found in literatures within the scope of this study. The results have been excellent and, in fact, have produced the best scores. Specifically, its correlation coefficient with the target variable was practically total (0.99). Likewise, the root-mean-square error (RMSE) between the experimental sensor data and the predicted variable was only 0.1351.

**Keywords:** hydrogen technology; PEM electrolyser; predictive maintenance; artificial intelligence; reinforcement learning; neural network; long short-term memory (LSTM)

## 1. Introduction

The hydrogen technology deployment is heavily dependent on cost-effectiveness. Regarding hydrogen-based microgrids, their economic impact is conditioned by costs of different nature (investment costs, operation and maintenance costs, and replacement costs) [1]. Among them, operation and maintenance are the ones that are repeated along the lifespan and affect the replacement costs. If equipment is not properly maintained, it arrives early at the end of its lifespan, and the equipment will need to be replaced much earlier.

In the case of electrolysers, they can only function effectively to produce hydrogen at the desired parameters if their components do not fail during the period of operation. Failures can be avoided if they are detected early during operation and resolved. The process of taking actions to monitor, detect, and resolve failures of a system is known as maintenance practice, as defined according to EN13306 [2]. Adequate maintenance of electrolysers will guarantee optimum operation at the designed level of efficiency, long-term cost-effectiveness, and safety.

There are various maintenance strategies of which the main categories are corrective maintenance, preventive maintenance, and predictive maintenance [3]. Corrective maintenance involves technical interventions carried out after a fault or damage has occurred, while preventive maintenance involves restorative actions performed at scheduled times to maintain the electrolyser in good condition. The challenge with preventive maintenance is that it does not consider the condition of the electrolyser before being deployed.

There are cases where the electrolyser is still in good condition and when maintenance is deployed too early, it results in a waste of resources such as labour and spares. Predictive maintenance was introduced as an advancement of preventive maintenance by predicting when failure will occur in the electrolyser. With the aid of such a predictive approach, labour, and spares are deployed just before the occurrence of faults [4]. The benefit of this type of maintenance strategy is that the electrolyser is able to operate for the designed lifespan. Predictive maintenance monitors the condition of the electrolyser using sensors embedded to capture data about its condition. The sensor data are analysed and used to predict imminent failure.

Various literature discussed predictive maintenance concepts as applied to electrolysers. In the study by Siracusano et al. [5], a PEM electrolyser was investigated using electrochemical techniques to individualise key degradation issues and predict MEA endurance under real-life operation. Also, Li et al. [6] investigated iron contaminations in feedwater for predictive maintenance considering that if not detected on time, can lead to accumulation both on the membrane and the catalytic layer, leading to increased ohmic and charge transfer resistance. Also [7,8] discussed the use of temperature in predictive maintenance to determine membrane drying and associated faults.

This paper presents a novel algorithm developed using Deep Reinforcement Learning (DRL) hybridised with a Long Short-term Memory (LSTM) neural network for artificial intelligent maintenance of an electrolyser. Figure 1 illustrates the focus of the paper.

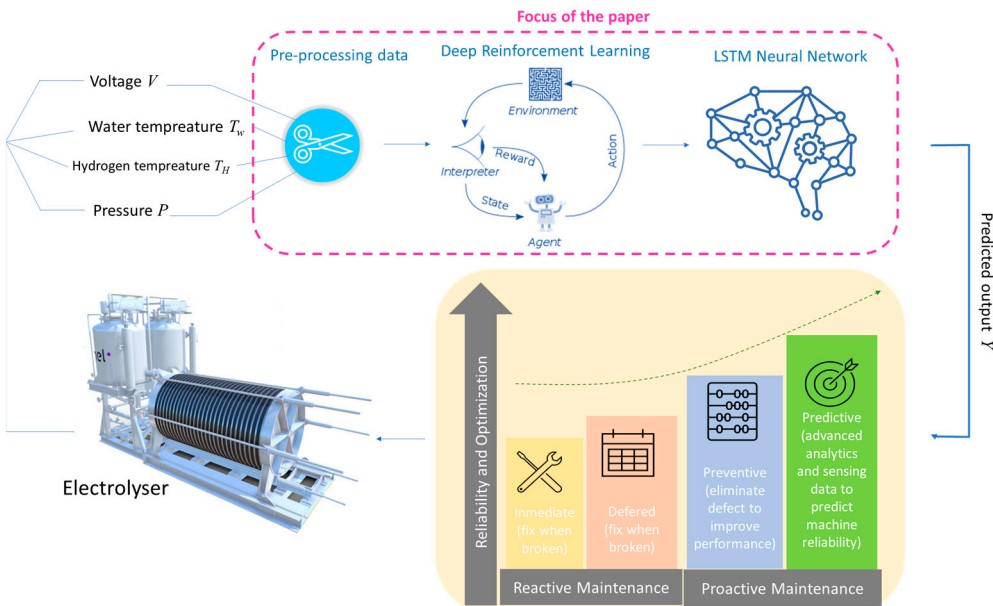

**Figure 1.** Illustration of problem statement and focus of the paper.

*Background and Previous Works*

The challenge with predictive maintenance is that sensors can fail thus giving false information to the predictive model used. Very limited studies exist within the field of intelligent maintenance of electrolysers that deal with this problem. To address the abovementioned issues, this study seeks to develop an intelligent predictive maintenance model to forecast sensor data in the electrolyser within some time frame. The model takes input data from other sensors for use in predicting a desired parameter which is referred to as the output of prediction. Hydrogen temperature is one such desired data based on its importance in proton exchange membrane (PEM) electrolysers as discussed in [8,9]. With the predicted data, an intelligent predictive model then allows monitoring and detection of any worsening condition of the electrolyser before failure.

Furthermore, to predict accurately, it is important that only data that have a strong correlation to the output data (hydrogen temperature) are selected, to ensure the accuracy

of prediction and reduction of computation time. Various input data that are captured by different sensors are referred to as features. Some of these features include pressure, voltage, cooling water temperature, and hydrogen temperature. The technique of selecting the correct feature (input) for use in accurate prediction is called feature selection [10].

According to previous works [10,11], there are various methods that can be used to perform feature selection which includes filters, wrappers, and embedded methods. Artificial intelligence has been used in various studies to perform feature selection such as carried out by Hazrati Fard et al. [10] and Kim et al. [12]. These papers used the reinforcement learning method to select the best features of a data set for prediction. The authors introduced algorithms to conduct an optimum search of features in the state space. An evaluation function is used to estimate the merit of each state according to selected features.

Regarding the scientific literature, very few authors focus their work on addressing the problem of maintenance in PEM electrolysers using artificial intelligence. For example, Kumar et al. [13] propose an artificial neural network based on LSTM capable of detecting and localising faults at every time step without any pre-processing. But the artificial intelligence-based faults detection system is only applied to power electronics; that is, the DC/DC converter that supplies the electrolyser, but the neural system is not able to detect faults in the electrolyser.

The works carried out by Mohamed et al. [14] are a clear example of using machine learning to predict the optimal design of PEM electrolytic cells. The machine learning-based model predicts up to eleven different parameters of the electrolytic cell using only four input parameters. In addition to the goodness of the model, it is not useful to detect faults or to define maintenance strategies.

But, as discussed earlier [1], to guarantee long-term cost-effectiveness and safety in electrolysers, it is necessary to define prognostics and health management (PHM) protocols that detect damages and helps to prevent catastrophic failure. With this premise, Lee et al. [15] present a PHM model based on machine learning to predict the load voltage of the electrolyser for the state of health information. The voltage is used as a state of health indicator which increases according to the time elapsed, and it is caused by the degradation of the electrolyser. Bahr et al. show in [16] the application of artificial neural networks in terms of modelling and simulating the aging process and the degradation of PEM electrolytic stacks. A broader model was developed by Zhao et al. in [17], where a data-driven digital-twin model is used to develop a dynamic model for predicting up to three variables (power consumption, hydrogen production, and temperature).

Based on the literature review carried out, the contributions of this paper are as follows:

(1)    A novel algorithm developed using DRL hybridised with LSTM. For predictive maintenance of an electrolyser, not all measured features can accurately predict the target sensor data, as the correlations between different features may be different. Any feature that is not well correlated with the data of interest will give an unassumable error during prediction. Hence, it is important to select appropriate feature(s) that, when used as input to the LSTM predictive model, will provide accurate prediction. In the case of multiple features obtained from the electrolyser, a manual process to select each feature(s) as input for the LSTM model followed by training with several parameter settings to reduce the prediction error is tedious. The developed DRL algorithm solves this problem by quickly searching through the feature set to select the one with the highest correlation to the sensor data (feature) of interest. A very important novelty of the DRL algorithm proposed in this paper lies in the unique method for evaluating each feature during iteration. The evaluation method is based on the comparison between a reference root-mean-square error (RMSE) and another one obtained from selected features. RMSE is an evaluation criterion used for LSTM models [18]. In the DRL algorithm, a simplified initial LSTM with one input and one output layer is used for evaluation. The best feature selected by the DRL algorithm is

the one with the lowest RMSE difference, which is then used as input to a main LSTM neural network to predict the sensor data of interest for predictive maintenance.

(2) Reduction of computational time when multiple features have to be evaluated. Another important novelty presented by the developed DRL algorithm is that when it has to select a feature set consisting of data from several sensors, it selects the average value, which gives a single representation of the combination. In addition, only a small sample of the feature set is needed. This approach saves computation time.

(3) Experimental and real-time operation. The algorithm takes information from experimental data and selects the best dataset (also called feature) that accurately predicts the sensor data of interest. This reinforces the advantages of the hybrid DRL-LSTM model for the maintenance of actual electrolysers.

(4) Optimisation of the parameters of the main LSTM prediction model. The parameters of the main LSTM neural network used to predict the target variable are optimised by keeping some constant and varying others to observe the effect on the RMSE. This study provides a graphical guide to selecting the optimal combination for accurate prediction with minimal adjustment and retraining.

(5) Digital twin approach. This study also demonstrates the digital twins approach to visualise electrolyser performance by comparing the actual sensor reading with the predicted output to identify electrolyser failures, allowing maintenance to be planned well in advance.

Table 1 summarises the main contributions of this study in comparison with other studies found in the scientific literature. The studies are classified based on key contributions and content relating to feature selection, evaluation, and deployment of artificial intelligence for the maintenance of electrolysers.

**Table 1.** Authors' contribution and related studies.

| Scientific Studies | Key Contributions | Data Selection and Evaluation Method | Artificial Intelligence Concept Used for Predictive Maintenance of Electrolyser |
|---|---|---|---|
| Authors' Proposal | Input data pre-processing. Predictive maintenance based on predicted data using artificial intelligence. | Novel algorithm based on hybridising DRL and LSTM neural network. | LSTM with optimised training strategy to reduce the root-mean-square error (RMSE). |
| Kumar et al. [13] | Open circuit fault prediction in the power converter of electrolyser. | n/a | LSTM |
| Mohamed et al. [14] | Prediction of different parameters of the electrolytic cell design using input parameters: hydrogen production rate, cathode area, anode area, and the type of cell design. | n/a | Machine learning models using polynomial and logistic regression. |
| Lee et al. [15] | Prognostics and health management model (PHM) to predict the load voltage of the electrolyser for the state of health information. The voltage is used as a state of health indicator which increases according to the time passed, and it is caused by the degradation of the electrolyser. | n/a | Machine learning models consisting of support vector machine (SVM) and Gaussian process regression (GPR) trained by using time, current, and power density as features and voltage as the output label to determine the potential fault. |

**Table 1.** *Cont.*

| Scientific Studies | Key Contributions | Data Selection and Evaluation Method | Artificial Intelligence Concept Used for Predictive Maintenance of Electrolyser |
| --- | --- | --- | --- |
| Bahr et al. [16] | Application of artificial neural networks (ANN) in terms of modelling and simulating the aging process. | n/a | ANN with Stochastic gradient descent (SGD) is used to find the optimal weight parameters for the desired relationship between input and outputs. Inputs to the neural network are stack current, temperature, and time while the output is the voltage. |
| Zhao et al. [17] | Data-driven digital-twin model to develop a dynamic model for predicting power consumption, hydrogen production, and temperature. | n/a | Fuzzy logic and neural network are implemented to improve the durability of the electrolyser. |

The remainder of the paper is organised as follows. Section 2 establishes the fundamentals of maintenance in electrolysers as well as the materials and methods applied by authors to develop a novel algorithm for feature selection using DRL. An LSTM neural network was then designed to use the selected feature for intelligent maintenance of a PEM electrolyser. Simulation results are shown in Section 3. Discussion of the results in Sections 4 and 5 provides the conclusions and future works.

## 2. Materials and Method

### 2.1. Initial Hypothesis

Green hydrogen is produced by electrolysis of water with no carbon emission. This process uses electricity to split water into hydrogen and oxygen. The electricity comes from renewable sources such as wind turbines, solar panels or hydropower [19]. PEM water electrolysis offers an efficient and flexible way to produce "green hydrogen" from renewable (intermittent) energy sources [19,20]. The schematics of a PEM electrolytic cell are shown in Figure 2.

Cell reactions:

$$\text{Anode}: \ 2\text{H}_2\text{O} \rightarrow 4\text{H}^+ + 4\text{e}^- + \text{O}_2 \tag{1}$$

$$\text{Cathode}: \ 2\text{H}^+ + 2\text{e}^- \rightarrow \text{H}_2 \tag{2}$$

$$\text{Overall}: \ 2\text{H}_2\text{O} \rightarrow \text{O}_2 + 2\text{H}_2 \tag{3}$$

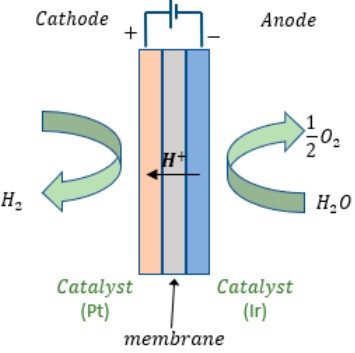

**Figure 2.** Schematics of a PEM electrolytic cell.

The equations governing the production of hydrogen are shown in (1) to (5).

$$\Delta G = \Delta H - T\Delta S \tag{4}$$

$$\dot{n}_{H2} = \frac{\eta_f n_{cel} i_{ely}}{zF} \tag{5}$$

where:

$\Delta G$ is the Gibbs free energy exchange (J).

$\Delta H$ is the enthalpy change (J).

$T$ is the reaction temperature (K).

$\Delta S$ is the entropy change ($JK^{-1}$).

$\dot{n}_{H2}$ is the molar flow rate of hydrogen (mol/h).

$\eta_f$ is the Faraday efficiency which is affected by temperature.

$n_{cel}$ is the number of cells.

$i_{ely}$ is the current through the electrolytic cell (A).

$z$ is the number of electrons in reduction reaction (2).

$F$ is the Faraday constant (26.81 Ah/mol).

One of the problems that affect reliability of a PEM electrolyser is membrane drying [9]. This is a function that is expressed in (6):

$$Drying\ effect\ of\ membrane = f(T,\ R,\ F_w,\ V,\ j) \tag{6}$$

where:

$T$ is the operating temperature (°C).

$R$ is the membrane resistance ($\Omega$).

$V$ is the voltage depending on the amount of current density (V).

$F_w$ is the condition of the feed water.

$j$ is the current density ($A/cm^2$).

In the study performed by Chandesris et al. [8], experimental results show that most of the membrane degradation occurs at the cathode side and also show the strong influence of the temperature on the degradation rate. In the study, the cell temperature is assumed uniform; hence, the temperature of hydrogen at the outlet of the cathode can be taken as the electrolyser temperature.

Hence, considering the importance of temperature in the degradation of electrolysers, in this paper, the authors use hydrogen temperature as the variable to be predicted for predictive maintenance in an electrolyser.

### 2.2. Materials for the Study

In a previous work [21], authors addressed the design, implementation, and control of the balance of plant (BoP) in a PEM electrolyser. Details of this previous work are shown in Figure 3.

From Figure 3, the sensor data used in the current study are those with features consisting of hydrogen pressure transmitter (PT112), hydrogen temperature transmitter (TT121), stack voltage (V), and cooling water temperature transmitter (TT105). Details of the technical characteristics of the electrolyser are shown in Table 2.

Experimental data showing some sensor data considered from the previous work are shown in Figure 4.

On the other hand, for this paper, we have used a computer equipped with an Intel(R) Core (TM) i5-8250U CPU @ 1.60 GHz 1.80 GHz, 8.00 GB memory, 64-bit, x64-based processor, Windows 11 Enterprise Operating System with MATLAB Programming environment (version R2022b).

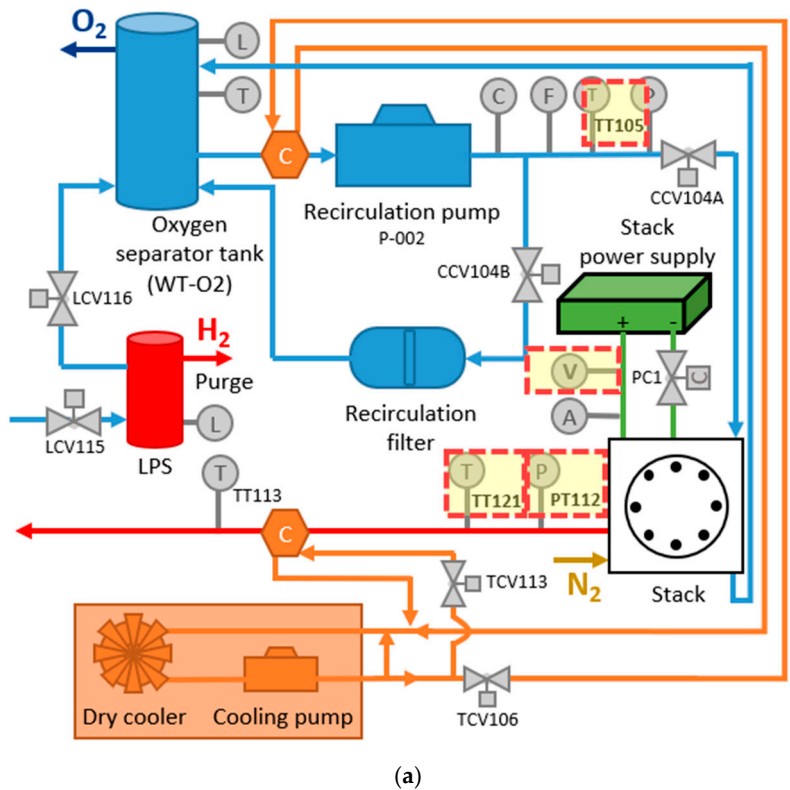

(**a**)

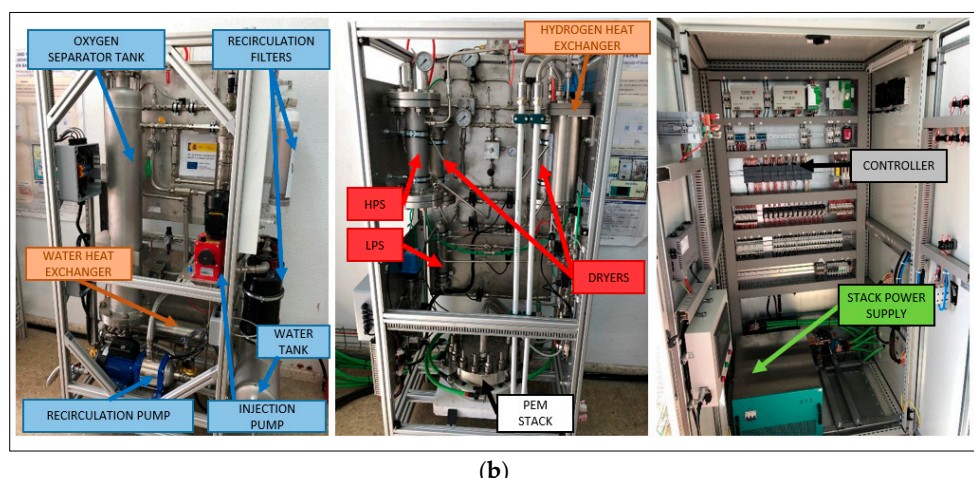

(**b**)

**Figure 3.** (**a**) Design of the BoP (sensors data highlighted as dashed squares are used in this study; (**b**) electrolyser implementation.

**Table 2.** Details of electrolyser used in this paper.

| Component | Stack Model | Technical Characteristics |
|---|---|---|
| PEM Electrolyser | GINER® Merrimack stack | $H_2$ production (Max): 2.22 Nm$^3$/h<br>Current density range: 300–3000 mA/cm$^2$<br>Maximum $H_2$ operating pressure: 40 bar<br>Maximum operating temperature: 70 °C<br>Cell voltage: 1.94 V<br>Cell dimensions: Ø 352.44 mm<br>Number of cells: 6 |

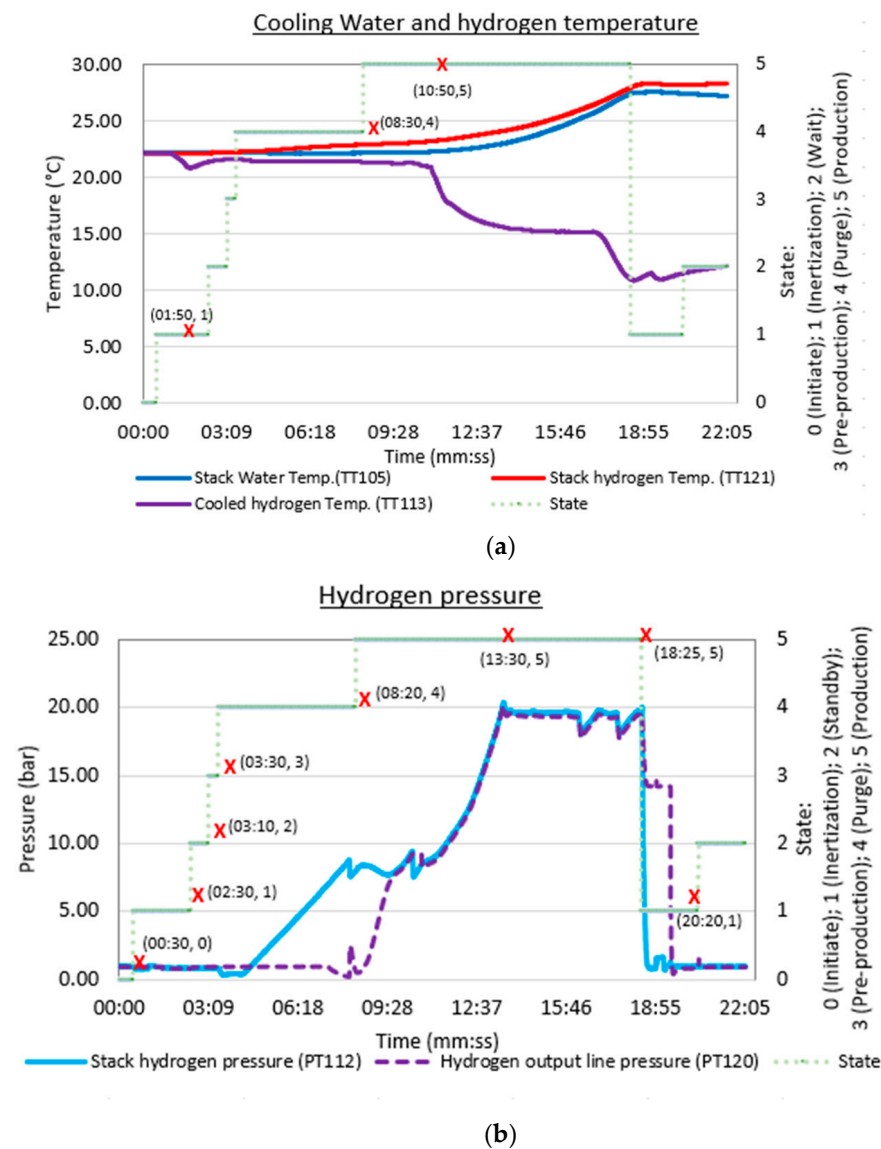

(**a**)

(**b**)

**Figure 4.** Experimental data from [21] showing profiles for (**a**) temperature and (**b**) pressure.

*2.3. Method Phase 1—Pre-Processing of Input Data*

The sensor data are merged into a single plot and superimposed based on the same time step as shown in Figure 5.

The lines labelled from a to h divide the time steps according to the operating mode of the electrolyser. During periods a to c, the electrolyser is in the initialising and waiting state. Data from sensor readings in this period are not changing, hence, such constant values cannot be used for prediction because it will negatively impact the accuracy of the output to be predicted [18]. Figure 5 shows that the data from time steps 500 to 2500, that is lines d to g, are appropriate to use for developing the predictive model with artificial intelligence. As mentioned earlier, hydrogen temperature (TT121) is the desired sensor data to be predicted as output, and its relationship with other sensor data (features) is shown in Figure 6.

A quick observation from Figure 6 is that it appears that the cooling water temperature (TT105) has a better correlation with the hydrogen temperature (TT121) which is the intended output to be predicted. However, there are other feature combinations that need to be explored to see which will give the best prediction of TT121. This process is called feature selection, as already discussed in Section 1.

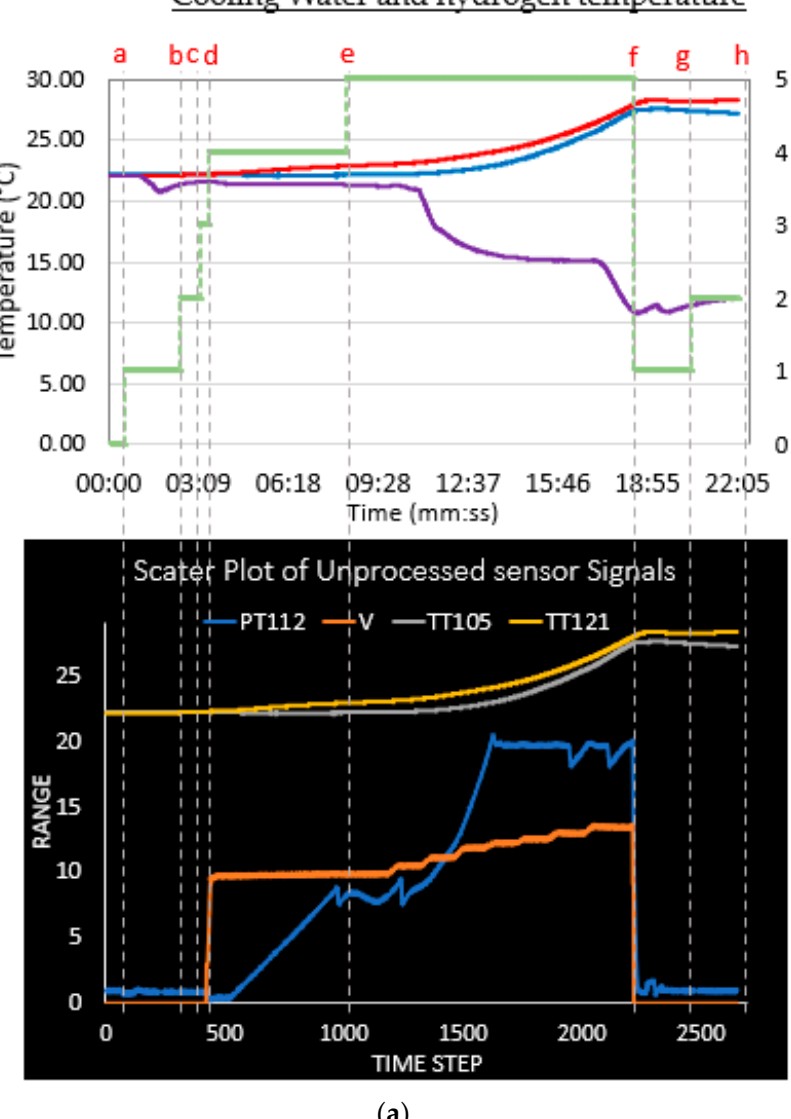

(**a**)

| Region | Activity being Preformed within the PEM Electrolyser |
|--------|------------------------------------------------------|
| a–h | Duration of Experiment |
| a–c | Initialization of the electrolyser |
| c–d | Pre-Production of hydrogen |
| d–e | Purging of the electrolyser |
| e–f | Full Production of hydrogen |
| f–g | Initialization |
| g–h | Waiting time |

(**b**)

**Figure 5.** (**a**) Pre-processing treatment of experimental data; (**b**) separation into regions.

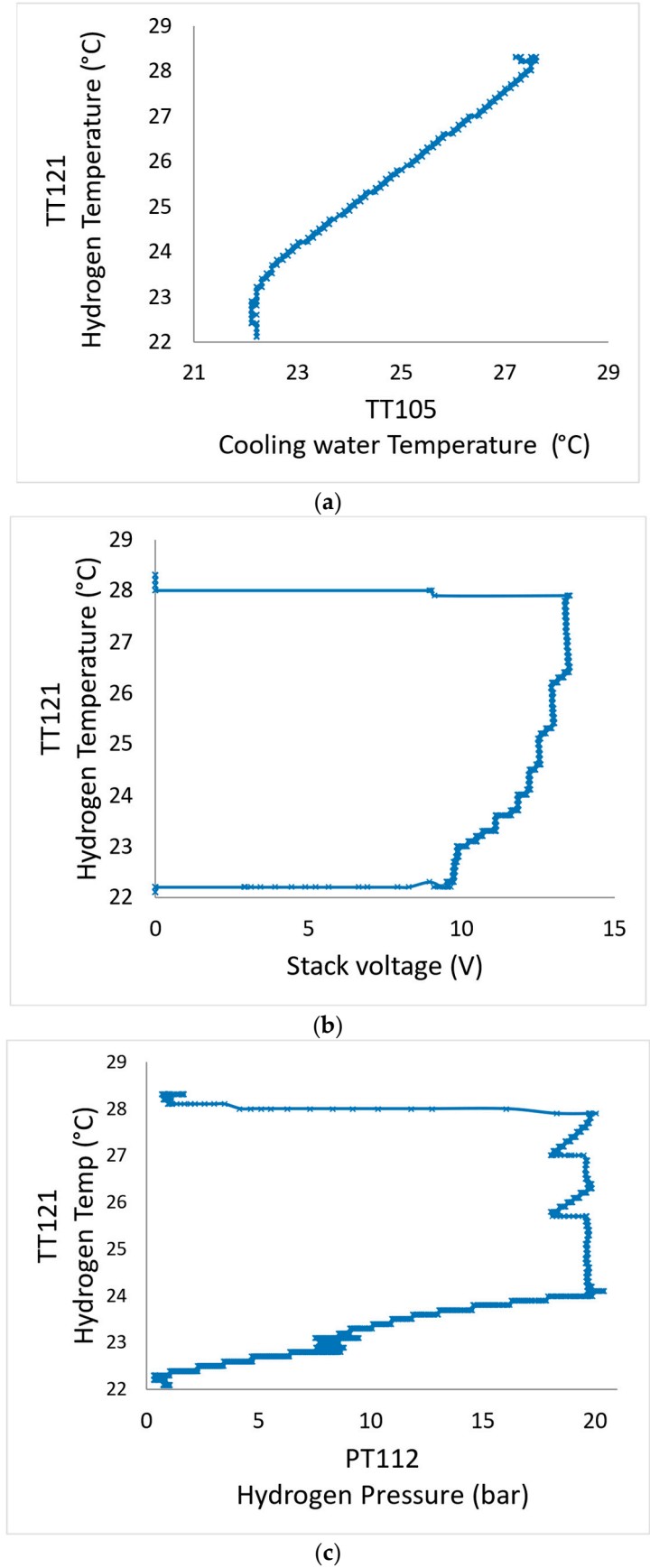

**Figure 6.** Relationship of hydrogen temperature with (**a**) stack water temperature; (**b**) stack voltage; and (**c**) hydrogen pressure.

For our study, the entire feature set that can be used for prediction is {[PT112], [TT105], [V], [PT112 TT105], [PT112 V], [TT105 V], [PT112 TT105 V]}. We are interested in determining which of these feature combinations will give the greatest prediction accuracy when fed into the intelligent predictive maintenance model that will be developed in the LSTM network.

A tedious and time-consuming process would be to take each subset of the features then feed it into the predictive model and check the root-mean-square error (RMSE). This manual approach can become even more complicated with features running into tens, hundreds or more from the dataset of electrolysers. To develop the algorithm to solve this feature selection problem, the experimental data obtained from sensors are pre-processed. Data pre-processing is necessary to be able to use them to predict accurately when using artificial neural networks. The first stage of pre-processing involves normalising or standardising the features to ensure that one feature will not affect the contribution of the other in the neural network [18].

The normalising process can be performed in several ways [16,22,23]. One alternative consists of ensuring that the range of all the features (data) is within 0 to 1 using the relation in (7). In another form called z-scoring, the input data are pre-processed using the relation in (8) where the data has a mean of 0 and a standard deviation of 1.

$$Normalization = \frac{Sensor\ Data(feature) - Min}{Max - Min} \tag{7}$$

$$Standardization(z-scoring) = \frac{Sensor\ Data(feature) - Mean}{Standard\ deviation} \tag{8}$$

Figure 7 shows a plot of pre-processed data using the methods of normalization and z-scoring. In this study, z-scoring is used considering its level of accuracy in previous work [18]. After pre-processing the data, a novel algorithm is developed based on deep reinforcement learning to select the feature that will be used as input to a main LSTM. The feature selected is the one that gives the best prediction of the output data with minimum error.

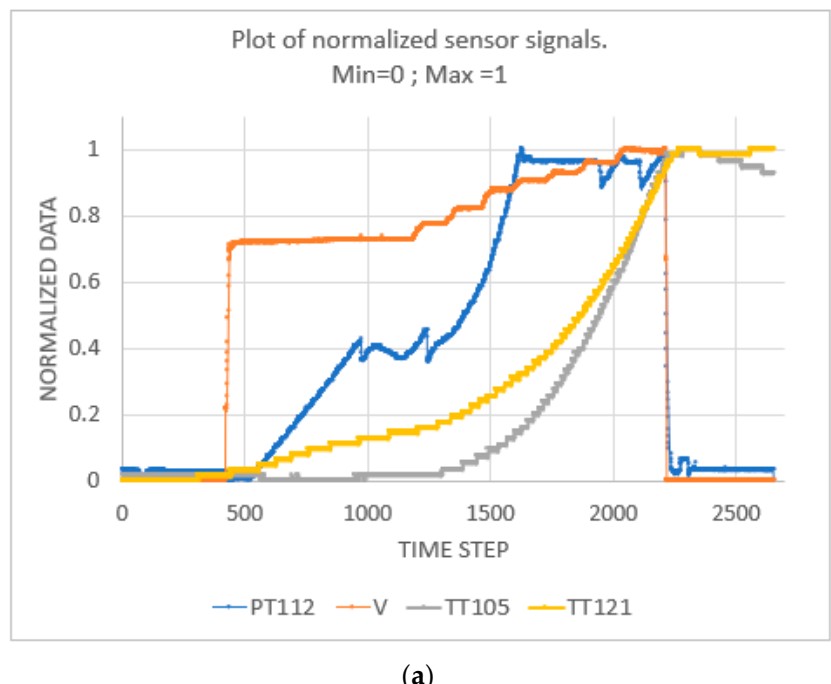

(**a**)

**Figure 7.** *Cont.*

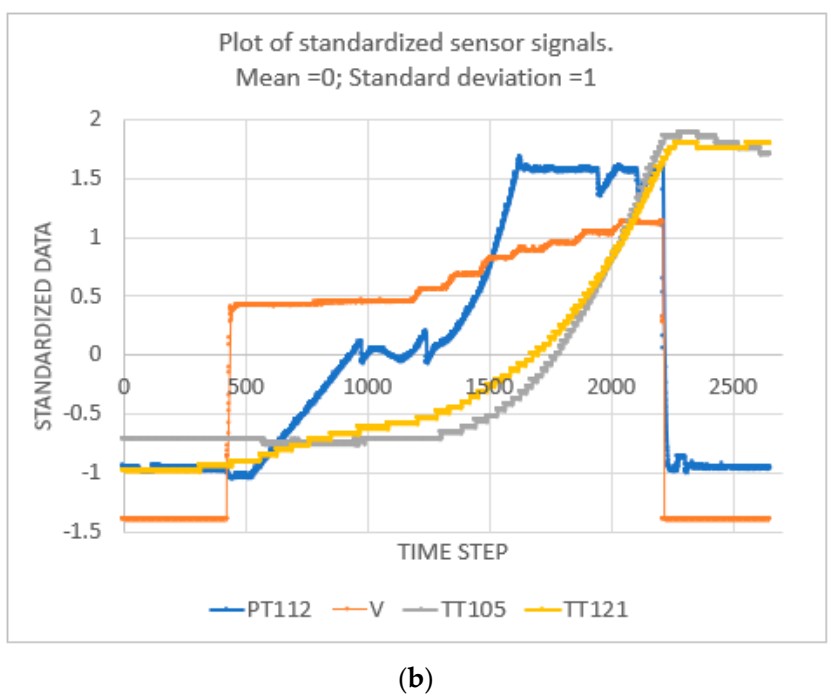

(**b**)

**Figure 7.** Plot of input features pre-processed using (**a**) normalisation and (**b**) z-scoring.

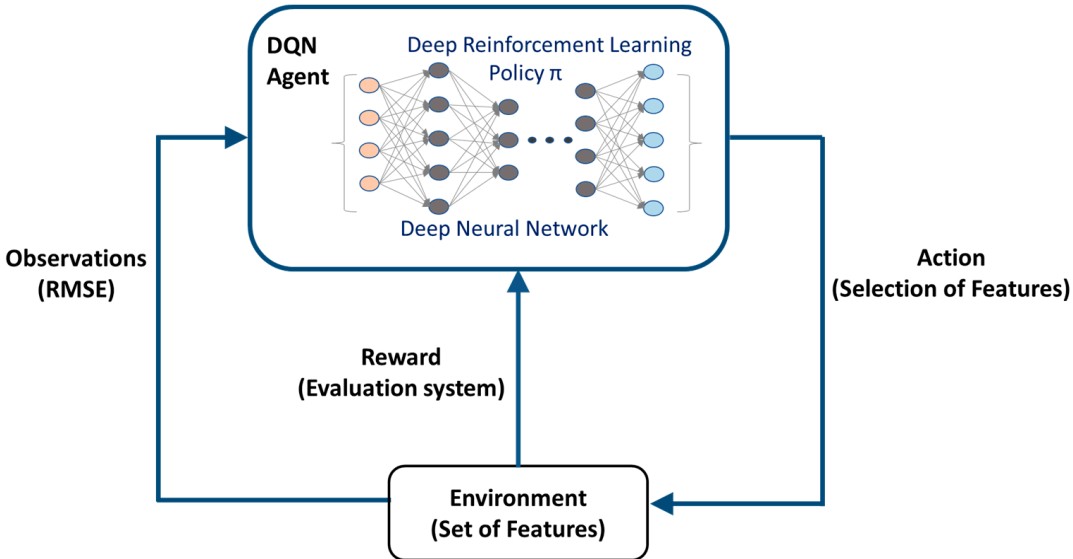

**Figure 8.** Schematics of DRL showing its agent.

For feature selection, the task of the agent will be to select the best subset of features that will give the most accurate prediction of the selected output (hydrogen temperature variable [TT121]). The whole set of features is the environment within which the agent operates while an algorithm within the agent evaluates the benefit of the feature selected. Before the agent begins to search through the features, it needs to be trained on how to recognise the actions that provide the maximum reward and others that take away the reward (penalise) when bad action is taken [10–12,24]. To build the algorithm for feature selection, we have considered the following:

**Environment**—It is the set of features {[PT112], [TT105], [V], [PT112 TT105], [PT112 V], [TT105 V], [PT112 TT105 V]} in a sample of data as shown in Figure 9.

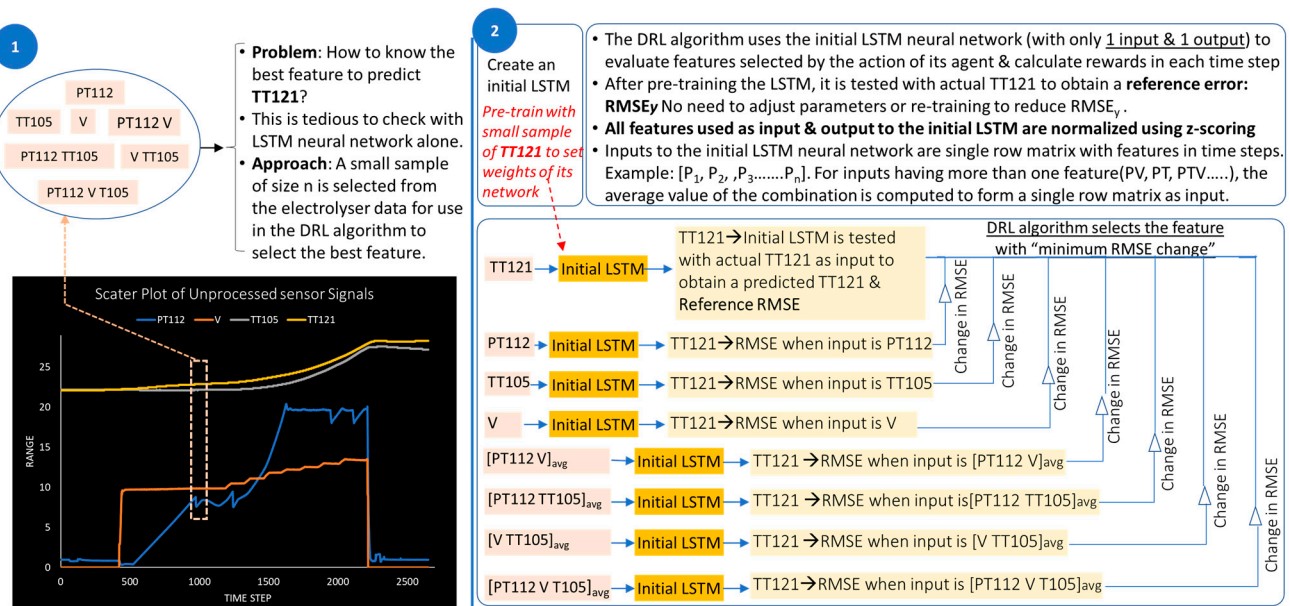

**Figure 9.** Schematics of methodology in the DRL algorithm for evaluating features to predict TT121.

**Action**—Selection of features to find out which is best to predict [TT121]. As there are 7 subsets of features, there will be 7 possible actions the agent can choose from in the space.

**Observation**—Each time an action is taken to select a feature subset. The observation generated from the environment is a $3 \times 1$ matrix consisting of the error of the previous action, current action, and integral of the current error (RMSE).

**Type of Agent based on Policy**—A Deep Q-Network (DQN) agent is used for taking action. It is a value-based reinforcement learning agent that trains a critic to estimate the return or future rewards [18]. The observations are continuous signals of errors while the actions are discrete (7 possible actions). During training, the agent does the following:

- Explores the action space of the set of features;
- During each iteration interval, the agent selects a random action for which the value function is the greatest;
- The agent updates a critic based on a mini-batch of experiences.

**Reward**—To determine the best next action in the environment consisting of a feature set, a measure is necessary to compare feature selection. Hazrati Fard et al. [10] introduced a criterion named AOR (average of rewards), which is the average evaluation of each feature whenever it is selected in the environment, and it is determined by (9).

$$AOR_f = Average\ [v(feature_t) - v(feature_{t+1})]\tag{9}$$

where:

$v(feature_t)$ is the assessment value function of the current state.

$v(feature_{t+1})$ is the assessment value function of the successor state.

We will use this approach to develop a unique evaluation system in our novel algorithm.

### 2.4. Method Phase 3—Authors' Approach—DRL-Based Algorithm for Feature Selection Using Novel Evaluation Criteria

Based on phase 1 and phase 2, the authors' approach consists of an algorithm that implements a DRL environment and a DQN agent with training parameters optimised based on Table 3. The reward for each action that the agent takes is determined by evaluating the minimum of an error difference (RMSE), as shown in expression (10).

$$RMSE\ gap = \min\left[RMSE_y(Prediction\ with\ T121) - RMSE_f(prediction\ of\ other\ features\ f)\right] \quad (10)$$

Expression (10) gives the minimum difference between the error (RMSE) arising from predicting TT121 by an initial LSTM neural network and another prediction of TT121 using the same (initial) LSTM when fed with each feature subset selected by the DRL agent. The methodology of the authors' approach is shown in Figure 9.

For the developed DRL algorithm to select features, only a small sample of the dataset is required after pre-processing which reduces computation time as well. Also, it is important to note from Figure 9 that for feature subsets consisting of more than one sensor data, an average value that gives a single representation of the combination is used for evaluation. Figure 10 shows an illustrative example of the methodology followed in the proposed DRL algorithm.

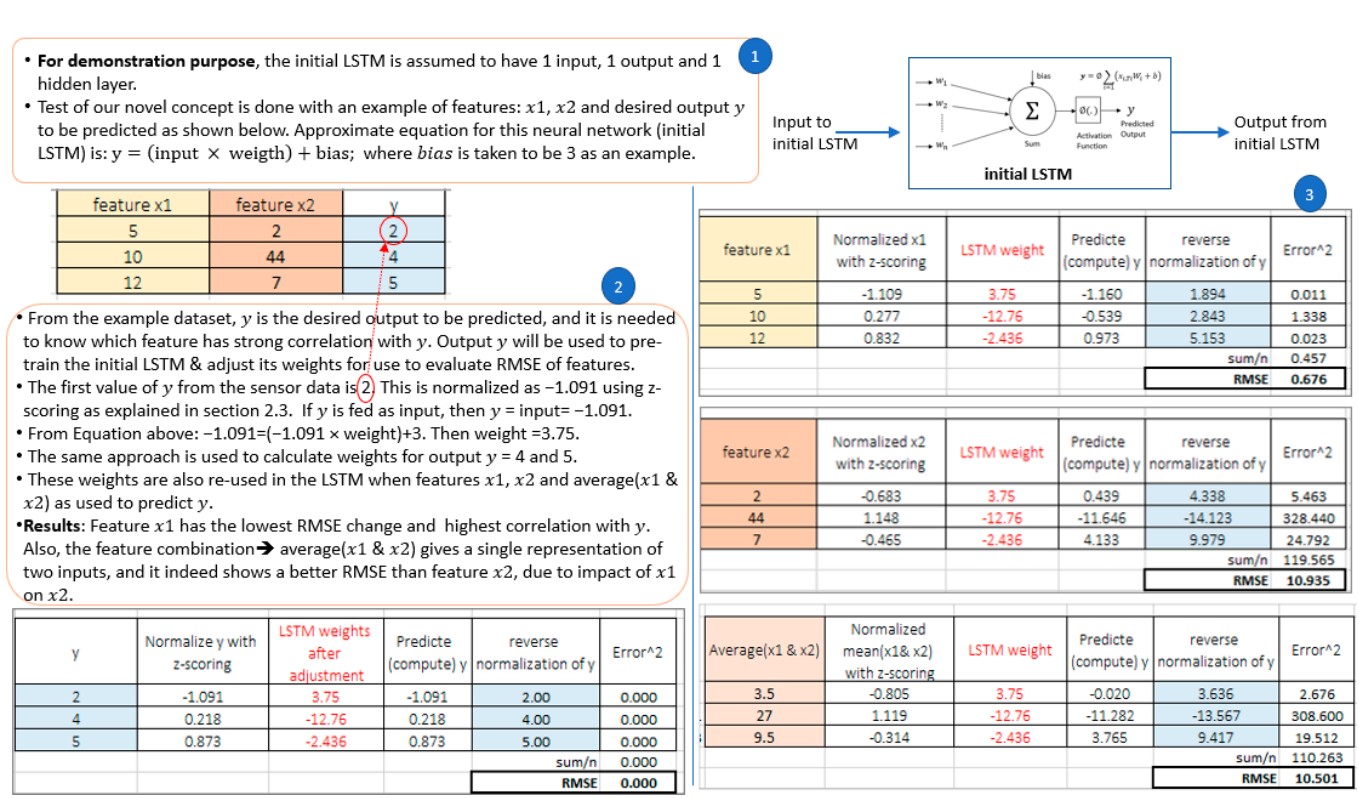

**Figure 10.** Illustrative example of feature selection by the developed DRL algorithm.

The single representation has inherited characteristics from each sensor data such as contribution to the RMSE. This approach makes the proposed DRL algorithm unique because it can handle complicated features subset of tens or hundreds of sensor data by reducing them to a single representation, which further reduces computation time.

Before the agent begins to take actions to select features, the algorithm checks that an appropriate weight has been computed in an initial LSTM neural network which has a simple architecture of 1 input and 1 output. It uses the matrix of TT121 which is our interested output ($y$) to train the LSTM and adjust its weights. Then, it is tested to obtain a reference $RMSE_y$, which is stored in memory. Thereafter, during iteration, the agent in the DRL algorithm (Algorithm 1) selects each feature subset from the matrix of features which is the environment. The pre-trained (initial) LSTM is then used again to compute another $RMSE_f$. The feature that gives the minimum difference in RMSE based on expression (10) is selected by the agent.

**Algorithm 1** Novel DRL algorithm proposed by authors

| | |
|---|---|
| ***Step 1:*** | *Input data consisting of features: pressure [PT112], cooling water temperature [TT105], stack voltage [V], and hydrogen temperature [TT121].* |
| ***Step 2:*** | *Normalise each feature using: z-scoring → (data-mean)/(standard_deviation).* |
| ***Step 3:*** | *Create and train an initial deep neural network (LTSM) with [TT121], as both input and output.* |
| ***Step 4:*** | *Obtain the matrix consisting of the subset of features.* *Subset 1 = [PT112]; Subset 2 = [TT105], Subset 3 = [V]; Subset 4 = [PT112, TT105], Subset 5 = [PT112, V]; Subset 6 = [TT105, V]; Subset 7 = [PT112, V, TT105].* |
| ***Step 5:*** | *For feature subsets having two or more sensor data, obtain a single representation by computing the average.* |
| ***Step 6:*** | *Create DRL environment model with observations and action settings based on* Table 3. |
| ***Step 7:*** | *Create a DRL agent based on DQN policy.* |
| ***Step 8:*** | *Define discrete actions for the DRL agent as scalar vector:* *Action = selection of features from the set in Step 4* |
| ***Step 9:*** | *Define the reward for the agent based on expression Equation (10):* *If action results in minimum RMSE difference* *Then reward = 2* *Else* *Reward = −1* *End if* |
| ***Step 10:*** | *Train the RL agent* *For: each stochastic action taken by the agent* *Receive observations from the environment model: error (RMSE)* *Calculate the reward of the selected action.* |
| ***Step 11:*** | *If reward > 0, then* *Store feature selected by the agent* *Else* *Take another action to select another feature from the subset* *Endif* *End* |

**Table 3.** DRL model information used in MATLAB.

| DRL Model Component | Type | Training Parameter | Value |
|---|---|---|---|
| DRL agent policy | DQN | Learning rate | 0.01 |
| | | Number of hidden layers | 128 |
| | | Gradient threshold | 1 |
| | | Discount factor | 0.99 |
| | | Batch size | 64 |
| | | Initial epsilon | 1 |
| | | Epsilon decay | 0.005 |
| | | Epsilon min | 0.01 |
| | | Number of training episodes | 50 |
| Environment | | Observation type | Continuous |
| | | Observation dimension | [3, 1] |
| | | Action type | Discrete |
| | | Actions | [1, 2, 3, 4, 5, 6, 7] |
| | | Observation lower limit | [−inf, −inf, 0] |
| | | Observation upper lower limit | [inf, inf, inf] |

Table 3 shows the training parameters of the DRL algorithm implemented.

The DRL algorithm is implemented in MATLAB Simulink as shown in Figure 11. The model consists of a trained DRL agent that is created as a block subsystem. The DRL model is trained with the parameters stated in Table 3 as required by the DQN agent and environment. The agent takes actions which is a scalar input from 1 to 7, based on training conducted earlier. The action is used as input to a MATLAB function which contains

the code developed by authors for interactions between the blocks. Another subsystem computes the observation matrix. There is also a subsystem to compute the reward based on expression (10).

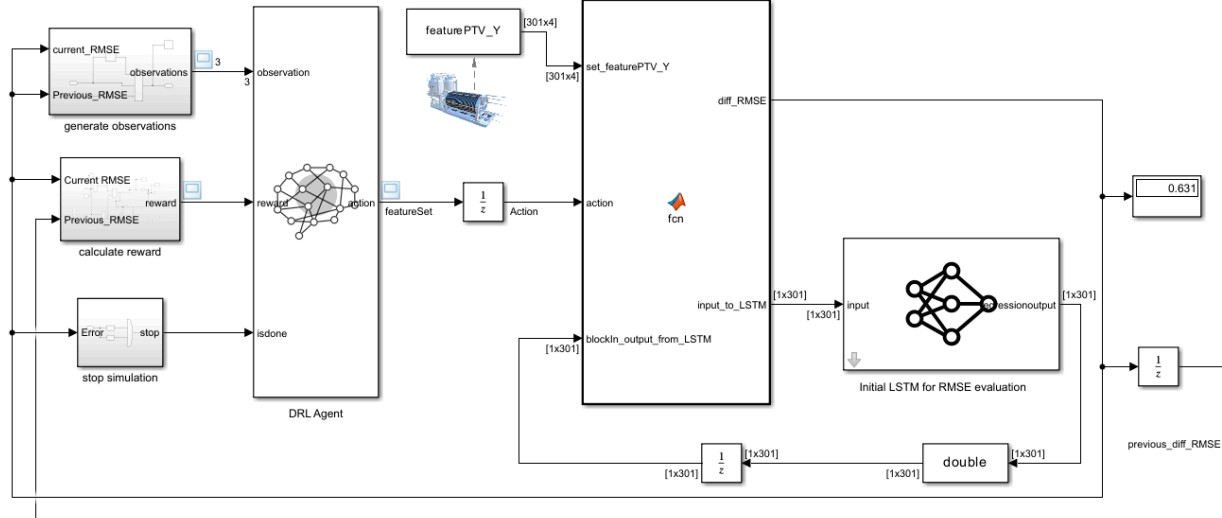

**Figure 11.** MATLAB programming environment for the developed DRL algorithm.

The reward is set up to have a scalar value of 2 when the agent takes an action that reduces the RMSE, while it is penalised by a scalar value of -1 when it takes an action that increases the RMSE.

Finally, there is a subsystem containing a simple deep neural network trained based on Figure 9, which we refer to as an initial LSTM. The internal weights have been adjusted when fed with variable TT121 and are ready for use to evaluate features with a strong correlation with TT121.

### 2.5. Method Phase 4—Artificial Intelligence-Based Predictive Maintenance for PEM Electrolyser

In this study, the artificial intelligence concept is further implemented with the aid of a main LSTM neural network, which will be used as the predictive model to forecast (predict) the hydrogen temperature (TT121) of the electrolyser within a time frame. The predictive model takes as input the best feature selected from a novel DRL algorithm developed by authors. The predicted output (TT121) will aid in planning maintenance for the electrolyser in the event that the hydrogen temperature sensor fails. This is even more important when the predicted temperature begins to rise above the optimum operating range of the electrolyser.

Based on the scientific literature [25], LSTM neural networks are a type of recurrent neural network (RNN) which itself is a subcategory under deep neural network (DNN). Then, DNN is a type of supervised learning under machine learning and artificial intelligence concepts. To alleviate the trouble of fading gradients in traditional RNN design, LSTM neural networks are configured from three gates, Figure 12, where they can store both present and historical information after it has been trained with input data.

According to Mikami [26], the relevant equation for each gate is as follows:

$$Forget\ Gate: f_t = \sigma\left(W_f.[h_{t-1}, x_t] + b_f\right) \tag{11}$$

$$Input\ Gate: i_t = \sigma\left(W_i.[h_{t-1}, x_t] + b_i\right) \tag{12}$$

$$Output\ Gate: O_t = \sigma\left(W_o.[h_{t-1}, x_t] + b_o\right) \tag{13}$$

$$Candidate\ valve\ C'_t = \tanh(W_c.[h_{t-1}, x_t] + b_c) \tag{14}$$

$$Cell\ State:\ C_t = f_t * C_{t-1} + i_t * C'_t \tag{15}$$

$$Hidden\ State:\ h_t = \sigma_t * \tanh(C_t) \tag{16}$$

where:

$t$ is the time step

$b$, is a bias added for each gate

$W_f$, $W_i$, and $W_o$ are the weight of each gate

$h_t$ and $h_{t-1,}$ are the output for the hidden layers in time steps t and $t - 1$, respectively

$x_t$, is the input at time $t$

$\sigma$ is the sigmoid activation function.

Based on these equations, our LSTM neural network can be designed in MATLAB using either a graphic interface or code by calling built-in libraries. This is shown in Figure 13.

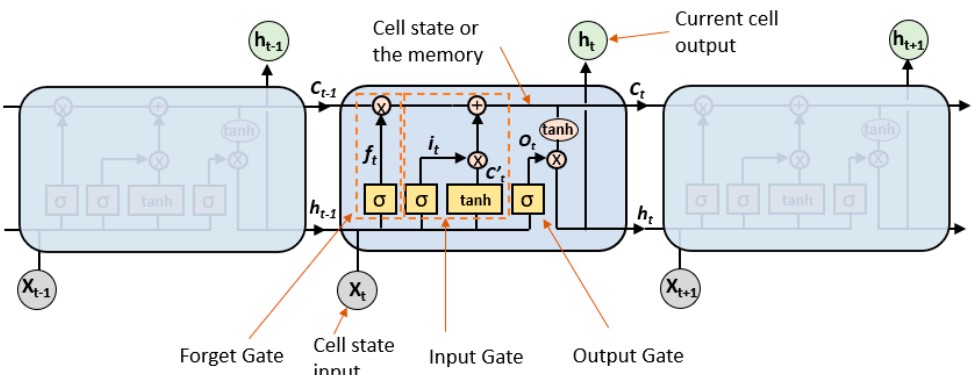

**Figure 12.** Conceptual design of LSTM neural network.

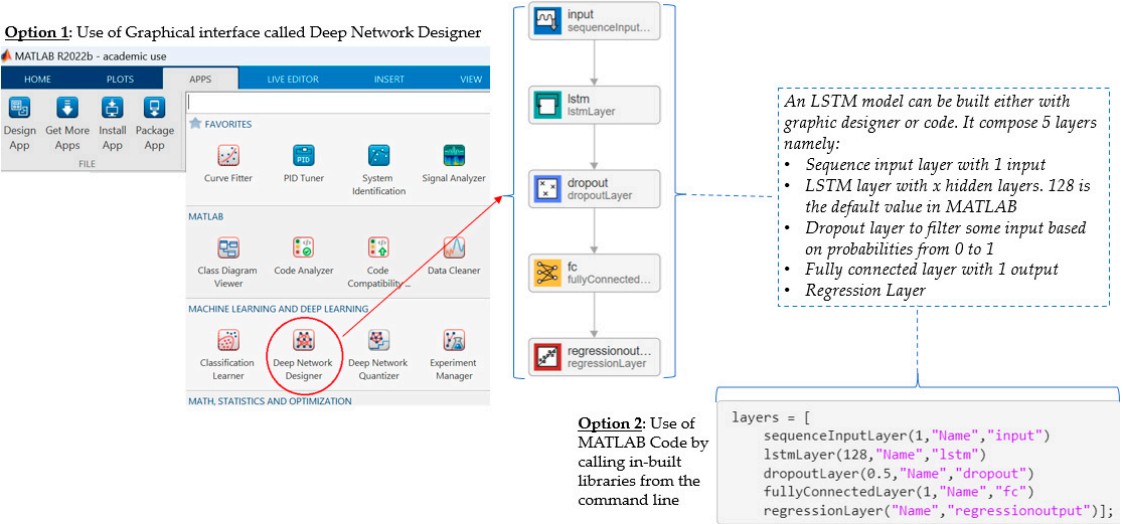

**Figure 13.** Design of LSTM in MATLAB.

MATLAB code is used to design the main LSTM and then to train it. The first training result is evaluated by checking the RMSE. If good enough, then the LSTM is ready for use in predicting the hydrogen temperature of the electrolyser. If not, the training parameters are adjusted again to obtain an optimum value of RMSE. The entire process is shown in Figure 14.

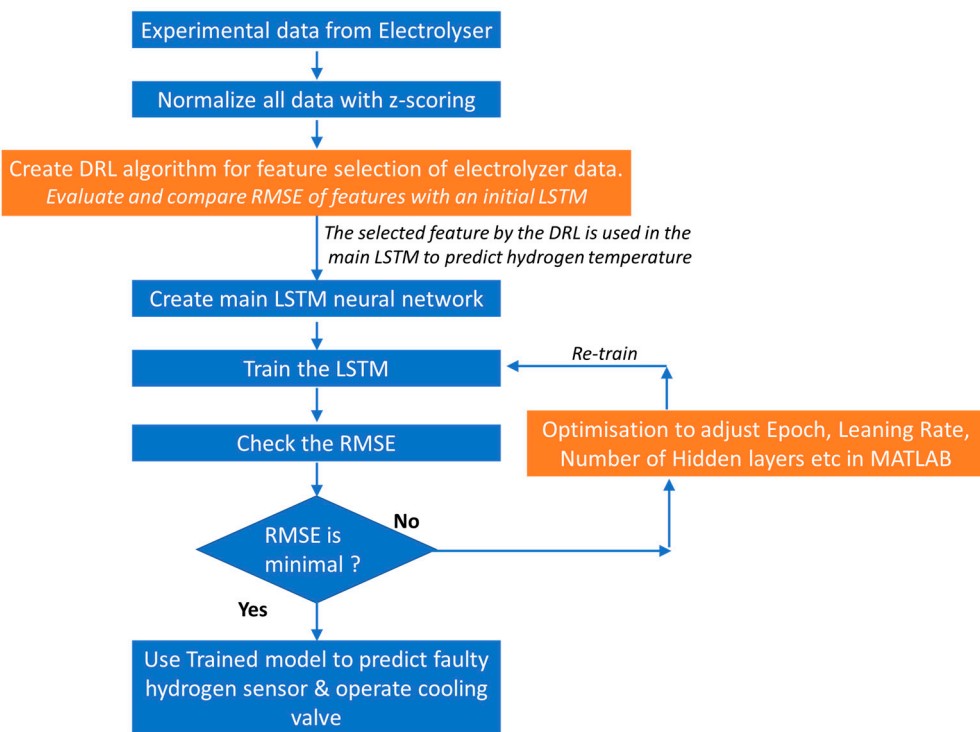

**Figure 14.** Flowchart for implementing intelligent maintenance in PEM electrolyser.

The overall layout showing the interaction between the DRL algorithm hybridised with LSTM neural network for use in the electrolyser control system is shown in Figure 15. The training parameters are adjusted to obtain an optimum value of RMSE, which gives the final prediction of the target variable, the hydrogen temperature [TT121].

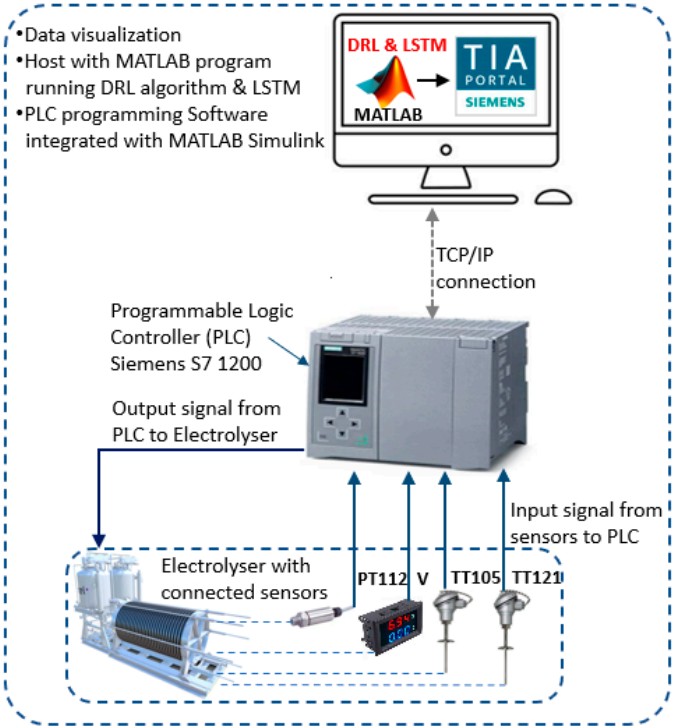

**Figure 15.** Overview of the implementation of intelligent maintenance in PEM electrolyser.

## 3. Results

This section will show the results of the feature selection obtained from the novel DRL algorithm developed by the authors, and, additionally, the results of training and prediction from the main LSTM neural network.

### 3.1. Results of Feature Selection Obtained from the Novel DRL Algorithm

The novel DRL algorithm described in Section 2 and implemented in MATLAB as shown in Figure 11, was trained for 50 episodes. The episode is the recording of actions, observations, and rewards for the DQN agent during training iteration. The training results offer an average reward of 77.8 as shown in Figure 16. Within the period of iterations, an average reward value above zero means that the agent took correct actions and accumulated 77.8 positive points. Any reward value below zero (negative) means the average action taken by the agent was wrong (recall Section 2.4).

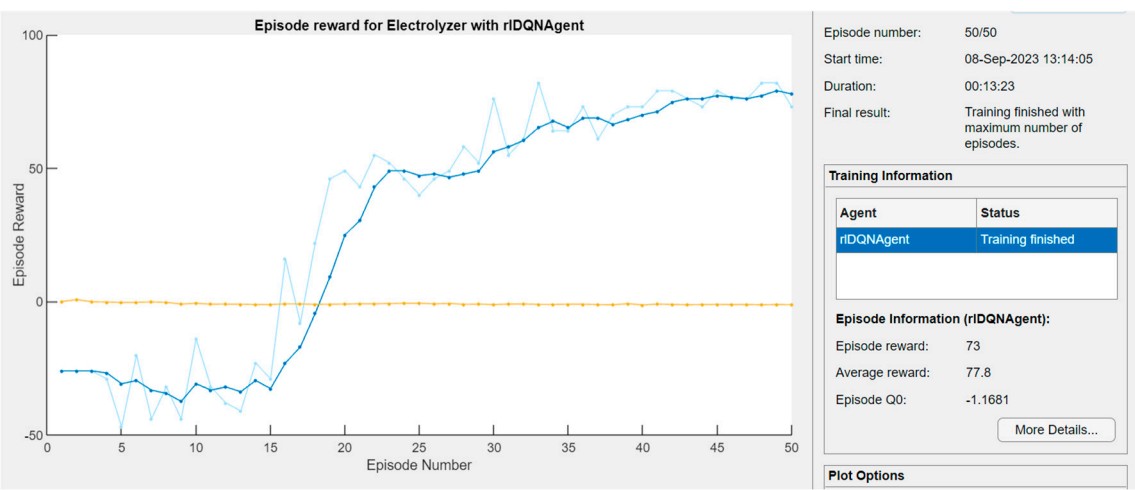

**Figure 16.** Training process of the DRL agent showing the rewards.

After the agent had been trained, it was tested to see which feature(s) fulfills the condition in expression Equation (10). Figure 17 shows the plot of the agent's action to select features and observations in terms of RMSE as well as the reward received for each action.

In Figure 17, it is important to note that there is a time-step difference between the instant when the agent takes actions and when observations are made. The observations are the computations for the RMSE difference based on expression Equation (10). At the start of the simulation, the agent selected feature subset 5, which is a subset of water temperature and voltage [TT105, V], and observed an RMSE difference (gap) of 4.04; hence, it received a reward value of −1. The next action taken by the agent is the selection of feature 1, which is a subset of hydrogen pressure [PT112]. The agent again received a reward of −1 because the RMSE gap was 3.8. However, when the agent selected the feature 2 subset, which is [TT105], it received a reward value of 2 since the RMSE gap was 0.631 and fulfills the condition Equation (10), indicating that it took the best feature. Hence, as shown in Figure 17, the trained DRL agent was able to select feature subset 2, [TT105], which is the cooling water temperature after an initial trial to pick feature subset 5 and 1, which are [PT112, V] and [PT112], respectively.

**Validation**: To validate the DRL agent's selection, a plot of the Pearson coefficient [27] was made in MATLAB to determine the relationship between the various features obtained from the electrolyser. In Figure 18, each diagonal plot contains the distribution of each variable as a histogram. For example, the first diagonal plot (1,1) contains the histogram distribution of variable [PT112]; the next one (2,2) contains the histogram variable TT105, and so on. Additionally, each off-diagonal plot contains a scatterplot of the variable indi-

cated on the left vertical axis with the variable indicated on the horizontal axis, including a pink least-squares reference line. That is, plot (1,2) shows the correlation scatterplot of variable PT112 with variable TT105. On the other hand, the number written in the left upper corner of each plot indicates the correlation coefficient. Thus, it can be seen that the variable TT121 (hydrogen temperature target variable) has the best correlation with stack water temperature [TT105] (correlation coefficient of 0.99). This matches the feature selected by the DRL algorithm.

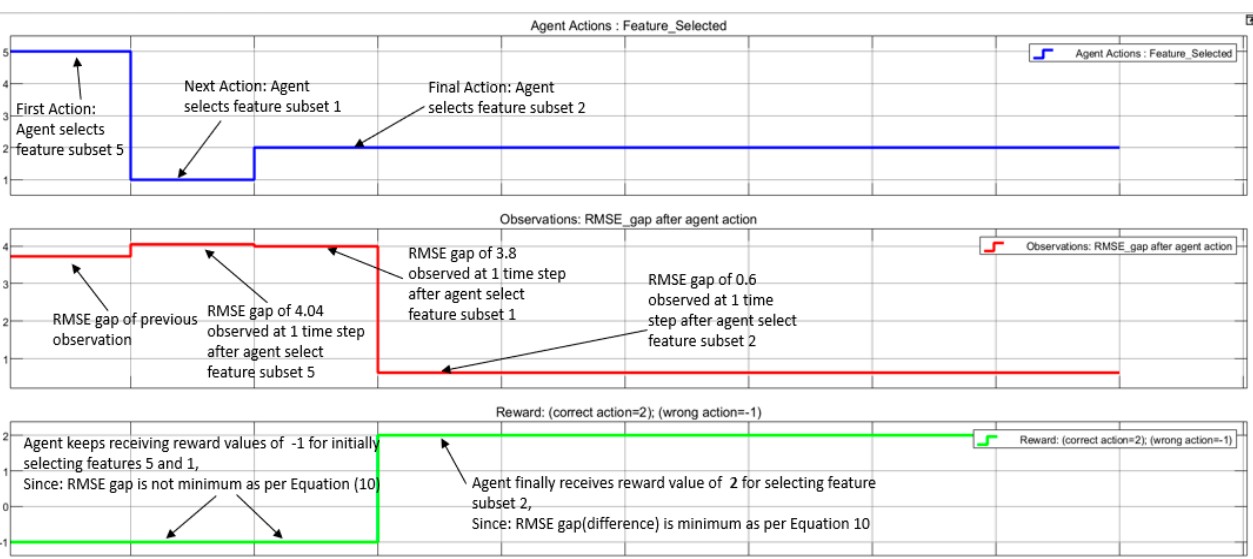

**Figure 17.** Plot from DRL model showing agent actions (in blue), current/previous RMSE (red), and rewards (green).

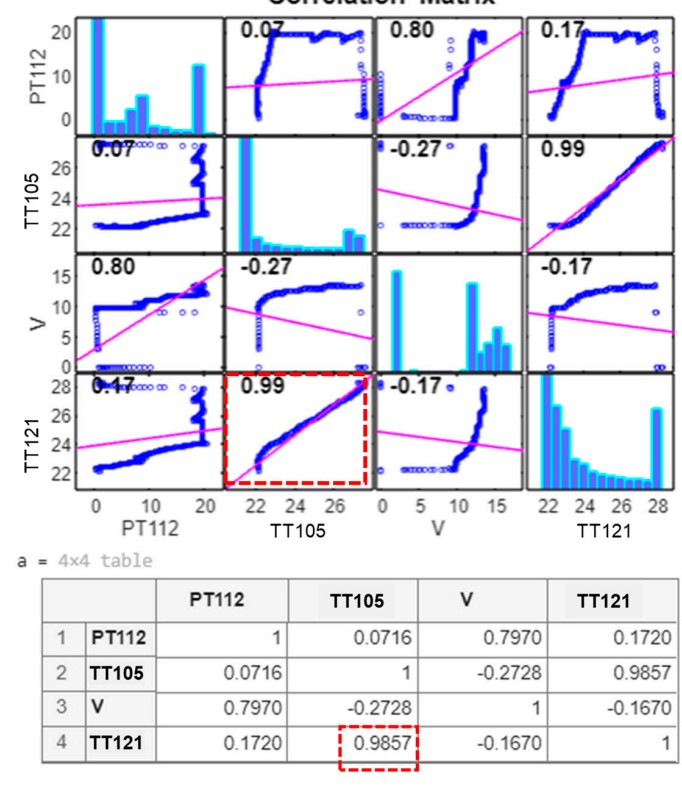

a = 4x4 table

|   |        | PT112  | TT105   | V       | TT121   |
|---|--------|--------|---------|---------|---------|
| 1 | PT112  | 1      | 0.0716  | 0.7970  | 0.1720  |
| 2 | TT105  | 0.0716 | 1       | -0.2728 | 0.9857  |
| 3 | V      | 0.7970 | -0.2728 | 1       | -0.1670 |
| 4 | TT121  | 0.1720 | 0.9857  | -0.1670 | 1       |

**Figure 18.** Pearson coefficient showing correlation between features (in red dashed lines the highest correlation).

### 3.2. Results of Training and Prediction (Testing) with the LSTM Neural Network

For the designed LSTM neural network, the data are divided into two parts as recommended in [18]. One part is 90% of the entire data and is used for training the network, while the remaining part (10%) is used to test the trained LSTM network for its accuracy in predicting the intended output [TT121], as follows:

- **Training data size**: time step 500:2300 (1800 data points = 90% of data);
- **Test data size**: time step 2301:2500 (200 data points = 10% of data).

There are several parameters that can influence the accuracy of the LSTM network such as the learning rate (LR), number of layers, epoch, gradient threshold, and dropout factor as described in [18]. Parametric analysis is performed to study these as shown in Figure 19.

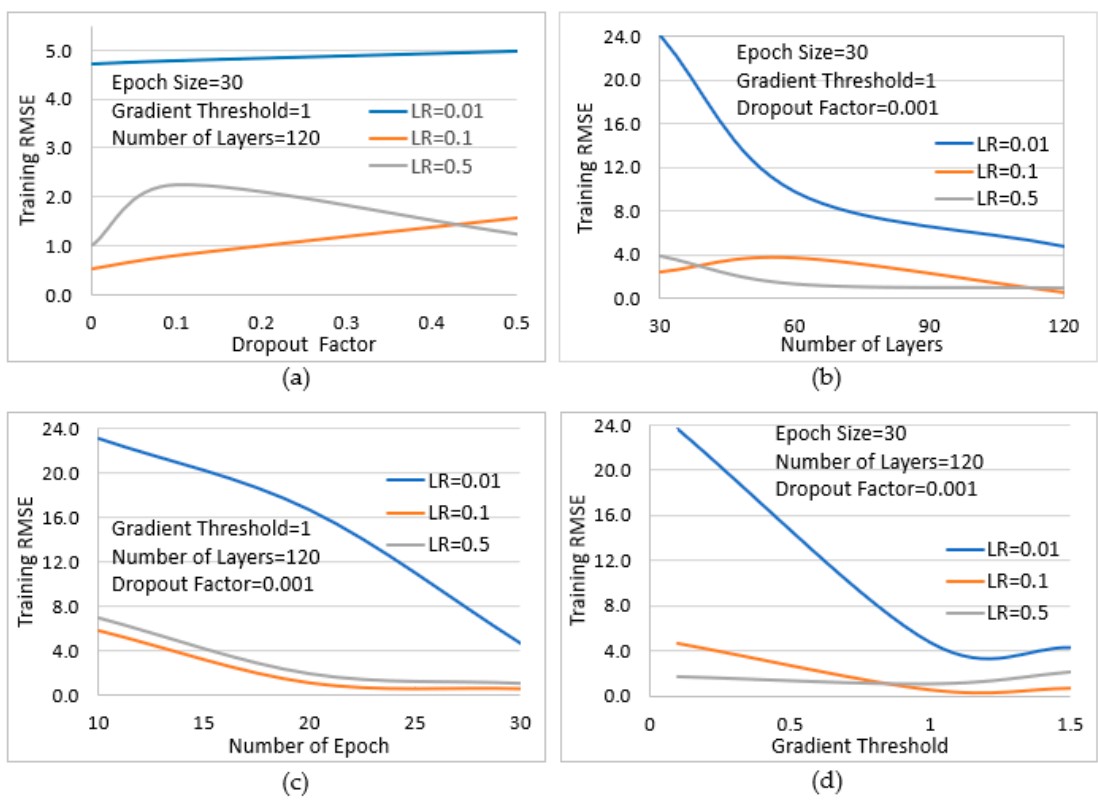

**Figure 19.** Optimisation of parameters to reduce RMSE during training of the LSTM for each selected learning rate (LR). The effect of variation in (**a**) dropout factor; (**b**) number of layers; (**c**) number of epochs; and (**d**) gradient threshold on the RMSE, are shown.

MATLAB programming environment was used to study the behaviour of the selected parameters by varying each one during training iteration while others were kept constant to determine the corresponding influence on RMSE. The RMSE obtained in this case is called training RMSE.

Figure 19 shows that a learning rate of 0.1 gives the lowest RMSE for all training iterations. Also, as the dropout factor approaches 0, the RMSE reduces significantly. In other cases, for the same learning rate of 0.1, a gradual increase in the number of layers and epoch value further reduces the RMSE. Finally, a gradient threshold of approximately 1 was seen to produce a much lower RMSE. From these observations, an optimum parameter set was obtained which resulted in a satisfactory RMSE of 0.09 during training. A plot of the training process is shown in Figure 20 while the paremeter set used for training are shown in Table 4.

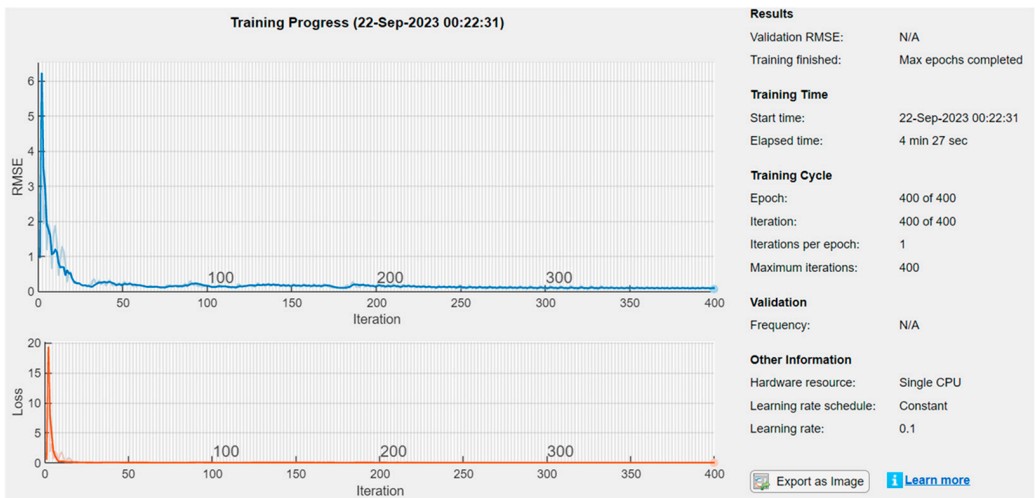

**Figure 20.** Final training result with training RMSE of 0.09.

**Table 4.** Final set of parameters used to reduce RMSE of the LSTM neural network.

| Training Parameter | Value | Training RMSE (LSTM) | Testing RMSE (Predicted Variable TT121) |
|---|---|---|---|
| Learning rate | 0.1 | | |
| Number of layers | 40 | | |
| Epoch | 400 | 0.09 | 0.1351 |
| Gradient threshold | 1 | | |
| Dropout | none (0) | | |

The trained LSTM network was tested by using it to predict the electrolyser hydrogen temperature ([TT121]). Satisfactory prediction was obtained as shown in Figure 21 with a testing RMSE of 0.1351.

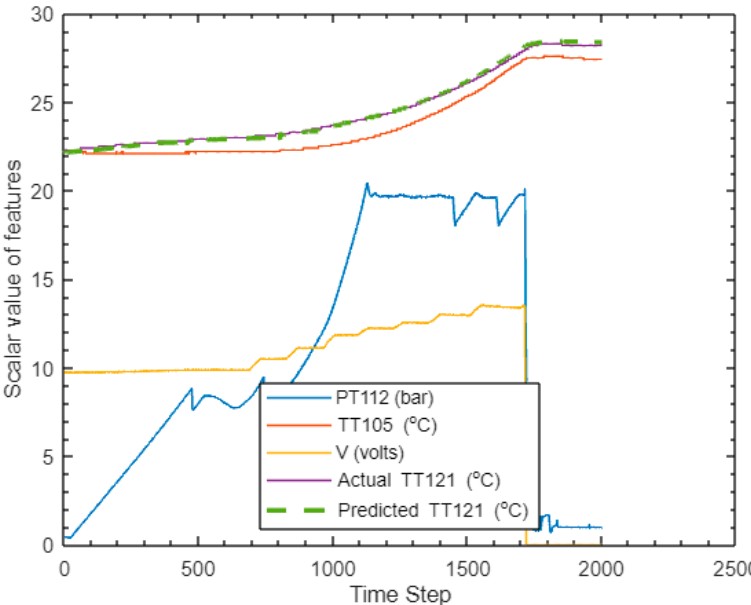

**Figure 21.** Target variable (hydrogen temperature TT121) predicted by developed LSTM neural network, and adjustment with experimental sensor data.

*3.3. Accuracy Validation—Comparison with Related Previous Works*

The RMSE obtained during the tests was compared with those corresponding to a few studies published in the field of reinforcement learning algorithms in the scope of this article. The results are shown in Table 5.

**Table 5.** Comparison of authors' proposal with other methods found in the literature.

| Study | Method Used | Testing RMSE |
|---|---|---|
| Authors' proposal | Hybrid of deep reinforcement learning (DRL) and long short-term memory (LSTM) | 0.1351 |
| Siraskar et al. [28] | Auto-setup reinforcement learning algorithm | 0.2082 |
| Duhirwe et al. [29] | Hybrid of DRL with extreme gradient boosting | 4.008 |
| Pannakkong et al. [30] | Reinforcement learning based on double DQN | 0.3956 * |
| Liu et al. [31] | Hybrid ensemble DRL | 0.9327 |
| Chen et al. [32] | Dynamic ensemble model based on deep reinforcement learning | 1.7416 |
| Almughram et al. [33] | Reinforcement learning hybridised with LSTM | 0.5196 * |

* RMSE is obtained from: $\sqrt{mean\ squared\ error\ (MSE)}$.

## 4. Discussion

In the novel DRL algorithm, authors used stack cooling water temperature transmitter data [TT121], to initially predict itself using a modified neural network which we refer to as an initial LSTM. The neural network's weight was adjusted to set a reference $RMSE_y$ for evaluation purposes, as shown in Figures 9 and 10. The obtained results demonstrated that the proposed DRL algorithm was able to select the best feature from the set of features obtained from the PEM electrolyser dataset. It is interesting to note that the algorithm used a small sample of data for the feature selection and incorporates a method of single (average) representation of features having multiple sensor data. Hence, it can be extended for cases with tens, hundreds or more feature subsets since it can reduce computation time.

The selected feature is the cooling water temperature [TT105] and was used in the LSTM neural network designed to predict hydrogen temperature. The selection made by the DRL agent was validated with the aid of the Pearson coefficient [27] which confirmed that the cooling water temperature [TT105] indeed has the highest correlation coefficient (0.99) with the target sensor (hydrogen temperature transmitter [TT121]), Figure 18.

The LSTM network was trained with optimised parameter set to achieve a training RMSE value of 0.09. The LTSM was then tested by predicting the hydrogen temperature at the outlet of the PEM electrolyser. The resulting prediction yielded a satisfactory accuracy with a testing (prediction) RMSE value of 0.1351 as shown in Table 4 and Figure 21.

The designed LSTM when deployed to HMI or SCADA systems can perform intelligent (predictive) maintenance of the PEM electrolyser using the accurately predicted output (in this case, hydrogen temperature). This solves the problem with predictive maintenance mentioned earlier, where failure of sensor data causes inaccuracies in predictive maintenance models. The model can also be beneficial when deployed on operator visualisation systems to provide a warning alert when the sensor fails. According to the previous work [21], the critical temperature level for hydrogen in the electrolyser considered for the study is 72 °C. The intelligent maintenance model can activate the cooling water system when the hydrogen temperature approaches this high value using the predicted data. If the faulty sensor is not replaced within a time frame, the developed system can also be useful to shut down the electrolyser for maintenance personnel to replace the faulty sensor.

Finally, Table 5 shows that the authors' proposal presents an excellent RMSE, fitting more accurately to the dataset than the other proposals found in the literature.

## 5. Conclusions

In this work, a novel DRL algorithm has been presented to apply artificial intelligence-based maintenance in electrolysers. The main contribution and novelty of the study lies in the DRL algorithm itself which is subsequently hybridised with a predictive LSTM model as follows:

- Novel DRL algorithm: This selects the feature with the highest correlation with the hydrogen temperature transmitter (TT121), whose data is to be predicted in a PEM electrolyser. One of the novelties of the DRL algorithm lies in the method of evaluating each selected feature by the action of its internal agent, as discussed in Section 2.4 and shown in Figures 8–10. The evaluation method is based on the comparison of a reference RMSE with the one obtained from the selected features. The DRL algorithm is also unique such that when the agent needs to select a feature subset consisting of more than one sensor data, the average value which gives a single representation of the combination is used for evaluation in the algorithm. This saves computation time.
- LSTM predictive model: The feature selected by the DRL algorithm is used by the LSTM model to predict the hydrogen temperature for use in predictive (smart) maintenance of the electrolyser. Parameters of the main LSTM were also studied using illustrative plots, as shown in Figure 19, to obtain an optimised combination to reduce the prediction error.

This study shows the benefits of reinforcement learning for the intelligent maintenance of renewable energy systems such as the electrolyser. This approach improves the durability of such equipment and ensures safety during the production of hydrogen, which is known to be highly inflammable.

The limitation of this study is that the DRL-LSTM hybrid model will need to be re-trained to adapt to a different hydrogen system, such as an alkaline electrolyser or fuel cell, whose physical phenomena are different from those for which it was trained in this study. Therefore, further work is needed to develop models that can be adapted to these different systems.

In addition, it is planned to implement the developed DRL-LSTM model (via TCP/IP communication) on the on-board devices of the electrolysers, such as the programmable logic controllers (PLC) and the human–machine interface (HMI), for maintenance purposes in the future.

**Author Contributions:** Conceptualization, A.A. and F.S.M.; methodology, A.A.; software, A.A.; validation, A.A. and F.S.M.; formal analysis, A.A., F.S.M. and J.M.A.; investigation, A.A., F.S.M. and J.M.A.; resources, A.A. and F.S.M.; data curation, A.A. and F.S.M.; writing—original draft preparation, A.A. and F.S.M.; writing—review and editing, F.S.M. and J.M.A.; supervision, J.M.A.; project administration, F.S.M.; funding acquisition, F.S.M. and J.M.A. All authors have read and agreed to the published version of the manuscript.

**Funding:** This research was funded by the Spanish Government, grant (1) Ref: PID2020-116616RB-C31 and grant (2) Ref: RED2022-134588-T REDGENERA.

**Data Availability Statement:** The data presented in this study are available on request from the corresponding authors. The data are not publicly available due to internal agreements between project partnerships.

**Conflicts of Interest:** The authors declare no conflict of interest.

## Nomenclature

| | |
|---|---|
| ANN | Artificial Neural Networks |
| $AOR_f$ | Average of Reward |



| BoP | Balance of Plant |
|---|---|
| DQN | Deep Q-network |
| DRL | Deep Reinforcement Learning |
| *F* | Condition of the feedwater |
| GPR | Gaussian Process Regression |
| HMI | Human Machine Interface |
| LSTM | Long Short-term Memory neural network |
| PEM | Proton Exchange Membrane |
| PHM | Prognostics and Health Management |
| RMSE | Root-mean-square error |
| MSE | Mean Squared Error |
| $RMSE_y$ | RMSE when the output $y$ of a feature set is used to predict itself |
| $RMSE_f$ | RMSE when a selected feature $f$ is used to predict the output |
| SCADA | Supervisory Control and Data Acquisition |
| SVM | Support Vector Machine |
| SGD | Stochastic Gradient Descent |
| LR | Learning Rate |

**Symbols**

| b | Bias added for each LSTM gate |
|---|---|
| $C_t$ | Cell state of the LSTM at time t |
| $C'_t$ | Candidate value of cell state in the LSTM |
| $\Delta G$ | Gibbs free energy exchange (J) |
| $\Delta H$ | Enthalpy change (J) |
| $\Delta S$ | Entropy change ($JK^{-1}$) |
| F | Faraday constant (26.81 Ah/mol) |
| $f_t$ | Forget gate of the LSTM |
| $h_t$ | Output for the hidden layer in LSTM at previous time $t$ |
| $h_{t-1}$ | Output for the hidden layer in LSTM at previous time $t-1$ |
| $i_{ely}$ | Current through the cell |
| $i_t$ | Input gate of the LSTM |
| $j$ | Current density |
| $n_{cell}$ | Number of cells |
| $n_f$ | Faraday efficiency which is affected by temperature |
| $\dot{n}_{H2}$ | Molar flow rate of hydrogen (mol/h) |
| $O_t$ | Output gate of the LSTM |
| PT112 | Hydrogen pressure (bar) |
| $R$ | Resistance of PEM membrane ($\Omega$) |
| $\sigma$ | Activation function (sigmoid function) in the LSTM |
| $T$ | Operating temperature of the electrolyser (°C) |
| TT105 | Cooling water temperature (°C) |
| TT121 | Hydrogen water temperature exiting electrolyser (°C) |
| V | Electrolyser voltage (V) |
| $v(feature_t)$ | Assessment value function of the current state |
| $v(feature_{t+1})$ | Assessment value function of successor states |
| $W_f$ | Weight of forgot gate in LSTM |
| $W_i$ | Weight of input gate in LSTM |
| $W_o$ | Weight of output gate in LSTM |
| $x_t$ | Input to the LSTM at time $t$ |

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
