# Peer review of "A Novel Deep Reinforcement Learning (DRL) Algorithm to Apply Artificial Intelligence-Based Maintenance in Electrolysers"

_algorithms, doi:10.3390/a16120541_

Round 1

Reviewer 1 Report

Comments and Suggestions for Authors

The study proposed a hybridizing deep reinforcement learning and LSTM-based artificial neural network, which works well. The article would be beneficial for predictive maintenance. However, some of the article's logical structure, methodology, and results are confusing. For predictive maintenance, more literature must emphasize some past practices. There are also few AI works of literature on DRL/DQN, making it difficult to emphasize its Novalty contribution. Here are some suggestions for the author's reference before it’s published. 

Generally, the DRL method will interact with a real environment and, of course, a physical model during pretraining. However the article uses a Pre-trained LSTM for DRL training. I don’t quite understand why a DRL architecture is needed if a Pre-trained LSTM can already predict successfully.

There is no mention of how DRL controls the hardware to improve it.

The architecture of DQN does not clearly explain what state, action, p, ..., etc, are operating.

In Figure 9, combining the input signal and the corresponding result to make predictions is illogical. What is being predicted by this approach? For example, supervised learning usually uses input signals to train neural networks to predict corresponding results. As shown in Figure 10, there is the same problem. It can also be organized to express x=[yi; ti].

Figure 11. Lack of interfaces for interacting with on-site hardware.

The core neural model used by DQN is not defined. The number of LSTM and hidden layers is not clearly defined.

The result and discussion, such as Figure 18, do not have a clear explanation.

Eq. (9) needs to define how to calculate the parameter v.

Sensor falls of electrolyzer falls? There is no description of how to make a difference in the experiment.

Reviewer 2 Report

Comments and Suggestions for Authors

1. The figure quality should be improved, such as Figure. 3.

2. The logical structure should be improved to make the proposed method more clearer to reader.

3. Some common Deep learning strategies are applied, however, the novelty is still not prominent.

4. The presentation of this paper needs to be improved.

Comments on the Quality of English Language

Moderate editing of English language required

Reviewer 3 Report

Comments and Suggestions for Authors

I have the following observations:

1.     This paper deal with A novel deep reinforcement learning (DRL) algorithm to apply artificial intelligence-2 based maintenance in electrolysers.

2.     Please give references at beginning of Definitions and of each new Equations set.

3.     Please compare your algorithm with others in terms of accuracy and other statistical quantities.

Round 2

Reviewer 1 Report

Comments and Suggestions for Authors

Thanks for your revising. I have no further questions.

Reviewer 2 Report

Comments and Suggestions for Authors

After reviewing the revised manuscript, we find that many problems still exists, for instance,

1) The tables, figures and texts are still blur and poorly presented.

2) The main results, such as Figure 18, are still unclear.

3) The novelty of this paper are required to be highlighted.

4) More comparison with other related methods should be presented.

5) Almost all the figures should be improved with high resolution.

Comments on the Quality of English Language

Moderate editing of English language required

Round 3

Reviewer 2 Report

Comments and Suggestions for Authors

1. The figures quality is poor, such as Figure 9, Figure 10, the images and texts are mixed together, which is difficult to read.

2. The resolution of related figures must be improved and carefully checked, such as Figure 18.

3. The main numerical results can be presented in the abstract.

4. Many clerical errors are appeared regarding the punctuation, spelling, sentences.

5. The logical structure should be improved.

6. What's the novelty of the proposed DL method compared with conventional DL method?

Comments on the Quality of English Language

Moderate editing of English language required

Round 4

Reviewer 2 Report

Comments and Suggestions for Authors

 The revised version has been improved a lot, but minor editing of English language and format according to the journal is required

Comments on the Quality of English Language

Minor editing of English language is required